# Childhood-Onset GH Deficiency versus Adult-Onset GH Deficiency: Relevant Differences Regarding Insulin Sensitivity

**DOI:** 10.3390/metabo12121251

**Published:** 2022-12-11

**Authors:** Heraldo Mendes Garmes, Alejandro Rosell Castillo, Sarah Monte Alegre, Aglecio Luiz de Souza, Yeelen Ballesteros Atala, Denise Engelbrecht Zantut-Wittmann

**Affiliations:** 1Endocrinology Division, Department of Internal Medicine, Faculty of Medical Sciences, State University of Campinas, Campinas 13083-852, SP, Brazil; 2Internal Medicine Division, Department of Internal Medicine, Faculty of Medical Sciences, State University of Campinas, Campinas 13083-852, SP, Brazil

**Keywords:** childhood-onset GH deficiency, euglycemic hyperinsulinemic clamp, insulin sensitivity, fat mass

## Abstract

The results of the studies on the pattern of insulin sensitivity (IS) are contradictory in patients with GH deficiency (GHD); however, the interference of the GHD onset stage, childhood or adulthood in the IS has not been assessed by euglycemic hyperinsulinemic clamp (EHC), a gold-standard method for the assessment of insulin sensitivity. In a prospective cross-sectional study, we assessed IS and body composition in 17 adults with hypopituitarism without GH replacement, ten with childhood-onset (COGHD) and seven with adulthood-onset (AOGHD) and compared them to paired control groups. COGHD presented higher IS (*p* = 0.0395) and a similar percentage of fat mass (PFM) to AOGHD. COGHD showed higher IS than the control group (0.0235), despite a higher PFM (0.0022). No differences were found between AODGH and the control group. In AOGHD and the control group, IS was negatively correlated with PFM (rs: −0.8214, *p* = 0.0234 and rs: −0.3639, *p* < 0.0344), while this correlation was not observed with COGHD (rs: −0.1152, *p* = 0.7514). Despite the higher PFM, patients with COGHD were more sensitive to insulin than paired healthy individuals, while patients with AOGHD showed similar IS compared to controls. The lack of GH early in life could modify the metabolic characteristics of tissues related to the glucose metabolism, inducing beneficial effects on IS that persist into adulthood. Thus, the glycometabolic findings in patients with COGHD seems to be not applicable to AOGHD.

## 1. Introduction

Growth hormone deficiency (GHD) is known to cause changes in body composition (BC), such as an increased fatty mass and decreased lean mass [1]. GH is a counter-regulatory hormone of the action of insulin and therefore, changes in glucose metabolism have been associated with GHD, such as hypoglycemia, especially in children [2]. Although the increase in fat mass found in these patients could justify a decrease in insulin sensitivity (IS) [3], the results of the studies on the pattern of insulin sensitivity are contradictory in patients with GHD [4]. Some authors have demonstrated a decrease in IS in patients with GHD [3,5], but other studies have even shown an increase [6,7]. Otherwise, studies in animals with extended longevity have shown greater IS in models with decreased prenatal GH action, despite an increased in percentage of fat mass (PFM) [4,8]. These results are similar to those found in most populations with a congenital decrease of GH action, as in patients with a mutation of the GHRH receptor gene and patients with Laron syndrome [9,10].

Many factors have been reported to explain these contradictory aspects about IS in GHD, such as the etiology of GHD and the influence of other hormone replacement therapies in patients with hypopituitarism [11,12]. Another important element is the method used to assess IS. In this sense, in a previous study, our group showed that, unlike what was observed in individuals with normal pituitary function, the agreement between the assessment of IS by the homeostatic model assessment for insulin resistance (HOMA-IR) and the euglycemic hyperinsulinemic clamp (EHC) was poor in patients with hypopituitarism [13]. This was also reported in another study where it is suggested that standard minimal homeostatic models to assess IS, such as HOMA-IR, could not adequately identify IS in syndromes of severe short stature [14].These findings highlighted the relevance of EHC in the investigation of IS, since the EHC does not suffer interference from the insulin counter-regulatory hormones, which are altered in patients with hypopituitarism [13]. On the other hand, the stage of the disease onset, childhood or adulthood, could influence IS in patients with GHD. While children with COGHD often present hypoglycemic events and are very sensitive to the insulin tolerance test, hypoglycemia is not a feature of GHD in adults [1,2]. However, to the best of our knowledge, no study has yet assessed the IS by EHC in adult patients with GHD according to the stage of disease onset, using the same hormone replacement protocol and comparing these groups of patients to healthy individuals paired by age, sex, and body mass index (BMI). For this reason, we decided to carry out a study to evaluate insulin sensitivity and body composition in adult patients with GH deficiency according to the patient’s age at the disease onset and compare it with a control group paired for age, sex, and BMI.

## 2. Materials and Methods

### 2.1. Study Design

This was a cross-sectional study. We evaluated adult patients with GHD and a previous diagnosis of hypopituitarism, all followed at a neuroendocrinology specialized center. All patients were on levothyroxine, prednisone, estrogen and progesterone or testosterone replacement and none were on GH therapy. Patients were divided into two groups according to the stage of the disease onset, childhood or adulthood. We assessed age, sex, and BMI, etiology of GHD, BC parameters, such as percentage of fat mass (PFM), percentage of free-fat mass (PFFM), fat mass weight, and free-fat mass weight (FFM) by means of electrical bioimpedance. Besides that, we assessed IS by EHC. The results of BC and IS of patients with GHD were compared with two control groups (CG) of healthy individuals paired for age, sex, and BMI for each group of patients with GHD. This clinical study complies with the Declaration of Helsinki and was approved by the ethics committee of Faculty of Medical Sciences, University of Campinas, UNICAMP (CAAAE 1.531.415) and all participants signed an informed consent form after a full explanation of the purpose and nature of all procedures used.

### 2.2. Patients

Seventeen patients between 20 and 59 years old with a diagnosis of hypopituitarism without GH replacement, were divided into two groups according to the age of disease onset, one group with ten adult patients with COGHD and another with seven adult patients with AOGHD. These patients were not under GH replacement, because this replacement in adult patients with GHD was not routinely performed in our outpatient clinic. Patients with COGHD were aged 6 (±1.16) years at diagnosis. All these patients received GH replacement, the mean treatment time was 10 (±1.11) years, and the treatment interruption time before the start of the study was 5 (±1.35) years. The diagnosis of pituitary dysfunction was accomplished in the presence of a deficiency of thyroid-stimulating hormone (TSH), free thyroxine (FT4), adrenocorticotropic hormone (ACTH), insulin-like growth factor (IGF-1), luteinizing hormone (LH), follicle stimulating hormone (FSH), total and free testosterone in male patients or estradiol in women. An insulin tolerance test was performed for the diagnosis of GHD and cortisol deficiency. All patients were under the replacement of levothyroxine (mean dose 1·56 mcg/kg/day), prednisone (mean dose 2·2 mg/m^2^/day), and estrogen/progesterone (17β estradiol 2 mg/day + norethisterone acetate 1 mg/day) or testosterone (testosterone cypionate 200 mg every three weeks), according to current guideline standards [15,16]. The only exception being a patient in the ACOGHD group who completed estrogen and progesterone replacement due to age. Adequate replacement was assumed when medication had not been adjusted for at least six months, patients had no complaints, and basal hormone levels were within the recommended values.

Exclusion criteria were the existence of a residual pituitary tumor or previous functioning tumor, diabetes mellitus, critically ill patients, malignant tumors, active inflammatory disease, class III/IV heart failure (NYHA classification), severe hepatic disease (low albumin or increased international normalized ratio), advanced kidney disease (stage four or five), HIV, or psychiatric diseases.

### 2.3. Control Groups

There were two groups of healthy individuals: one group consisted of twenty and the other of fourteen individuals, both of which were paired for age, gender, and BMI, with the groups of patients with GHD. The healthy individuals were recruited among the accompanist of patients in the outpatient clinic of our hospital. The IS and body composition characteristics evaluated in the group of adult patients with GHD were assessed in the control groups in the same laboratory and by the same researchers. All patients and control subjects included in this study were of Caucasoid ethnicity

### 2.4. Clinical and Body Composition Parameters

Clinical data were recorded consisting of age, sex, BMI, and etiology of GHD. BMI was calculated based on the ratio between the body mass (in kg) and squared height (in meters). In all of the study subjects, BC variables were evaluated by electrical bioimpedance with a Biodynamics monitor (Biodynamics Corp., Seattle, WA, USA), as described by Boulier A et al. [17]. The measurements were taken in the morning after each subject had fasted at least ten hours and their bladder was voided. The subjects had been supine for at least five min, arms relaxed at the sides but not touching the body and thighs separated.

### 2.5. Laboratory Measurements

Plasma glucose was measured with the glucose oxidase method using an YSI glucose analyzer (YSI 2300-Stat Plus analyzer; YSI, Yellow Springs, OH, USA; RV: 70–100 mg/dL).

### 2.6. Euglycemic Hyperinsulinemic Clamp Study

The EHC clamp study, which was carried out after an overnight fast, consisted of two hours of euglycemic insulin infusion at a rate of 40 μU/min per meter squared of body surface area and was preceded by two hours of a control period, as previously described [18]. A polyethylene, 20-gauge catheter was inserted into an antecubital vein for the infusion of insulin and glucose. Another catheter was inserted retrograde into a wrist vein, and the hand was placed in a heated box (50 °C–60 °C) for the sampling of arterialized blood. Following this procedure, the patients rested for at least 30 min in the supine position. The infusion was adjusted according to glucose determinations made every five min on a glucose analyzer (Yellow Springs Instruments, Yellow Springs, OH, USA). For the calculation of IS from the glucose disposal rate (M-value), milligrams per kilogram per minute was carried out using the infusion rate of exogenous glucose during the second hour of the insulin clamp period, with the M-value normalized per kg FFM.

### 2.7. Statistical Analysis

The frequency tables of the categorical variables with absolute (n) and percentage (%) values were used to describe the profile of the sample according to the variables under study. Descriptive statistics of the numerical variables, with values of mean, standard deviation, minimum, maximum, and median values were performed. Categorical variables were compared using the Chi-square test and, when necessary, the Fisher exact test. The Mann–Whitney test was used to compare numerical variables. In the comparative analysis between the two groups of patients with GHD, the variables evaluated were adjusted for age and BMI. In the correlation analysis to evaluate the relationship between M-value/kg of free fat mass (M/FFM) and PFM values, the Spearman correlation coefficient was used. The confidence interval (CI) was 95%. The level of significance adopted for the study was 5%.

## 3. Results

### 3.1. Clinical Characteristics of Adult Patients with GHD without GH Replacement

Our study included 10 patients with COGHD and 7 patients with AOGHD. The description of each patient in two groups regarding age, sex, BMI, treatment, etiology, IGF-1, and fasting glycemic levels, body composition variables, such as PFFM and PFM, as well as M/FFM are shown in Table 1.

### 3.2. Comparative Analysis between COGHD and AOGHD Patients

Regarding IS, we found higher M/FFM values among patients with COGHD compared to patients with AOGHD, these values were adjusted for age and BMI. These groups did not show differences in sex, IGF-1, and fasting glucose levels; however they showed differences in age and BMI. PFM and PFFM were similar between these groups (Table 2 and Figure 1).

### 3.3. Comparative Analysis between COGHD Patients and Respective Control Group

These groups did not present differences in age, sex, and BMI. Fasting glucose and IGF-1 levels were lower in patients with COGHD than in controls. When body composition was analyzed, patients with COGHD had a higher PFM and lower PFFM, compared with the control group (Table 2). Regarding IS, we found higher M/FFM values among patients with COGHD compared to the individuals of the control group (Figure 1 and Figure 2).

### 3.4. Comparative Analysis between AOGHD Patients and Respective Control Group

These groups did not show differences in age, sex, BMI, fasting glycemic levels, as well as body composition parameters. Only levels of IGF-1 were significantly lower in patients with AOGHD (Table 2). At the same time, M/FFM values were not different between these groups showing similar IS (Figure 1).

### 3.5. Correlation between IS and PFM in Adult Patients with GHD according to the Stage of the Disease Onset and Control Group

Control group and AOGHD patients showed a negative correlation between IS and PFM; however, we did not find this correlation in the patients with COGHD (Figure 3).

## 4. Discussion

It is known that insulin resistance is related to type 2 diabetes and obesity, diseases that increase cardiocirculatory mortality. Understanding the IS in adult patients with GHD is important for the management of these patients. In this sense, we assessed the IS in COGHD and AOGHD and found that IS was higher in patients with COGHD than AOGHD, even when presenting a similar body composition. In addition, despite lower PFFM and higher PFM values, IS was higher in patients with COGHD than in a paired control group. On the other hand, patients with AOGHD did not show IS differences in relation to a control group.

It is noteworthy that, initially, our patients were selected for an evaluation of IS, regardless of the patient’s age at the disease onset for a previously published study [13]. Subsequently, it was decided to divide them into two groups: COGHD and AOGHD, groups not matched for BMI and age. The direct comparison between these two different groups is not ideal, but the influence of the lower BMI and age found in patients with COGHD on IS results was statistically compensated with adjustments for age and BMI.

Furthermore, it is known that PFM is more important to determine IS or cardiovascular risk than the BMI [19,20] and it is worth mentioning that, in the present study, PFM was similar in both groups of patients. To corroborate that patients with COGHD were more sensitive to insulin independently from BMI and age, we compared these two groups of patients with a respective control group paired by BMI, sex, and age and we found higher IS in patients with COGHD than in their control group, while patients with AOGHD had similar IS to their respective control group, showing that the stage of GHD onset may be a factor that influences IS. We highlight that a study using an insulin tolerance test (ITT) has already shown that young GH deficient children, despite being heavier and fatter, were more insulin sensitive than children with normal GH secretion [21].

The results of studies evaluating IS in patients with GHD are controversial [4]. Most studies do not separate patients with COGHD and AOGHD. Although several authors associate GHD in adults with insulin resistance, this association is not well established in the literature. Conversely, a recent study has shown that adult patients with GHD are more sensitive to insulin than the control group matched for age, sex, and BMI through HOMA-IR [22,23]. In our study, using the HEC, we found no differences in SI between patients with AOGHD and the matched control group.

The reason why patients with COGHD have higher IS despite having a greater PFM than paired controls needs to be better understood. It is probable that GH deficiency installed since childhood facilitates some specific functional changes, inducing the development of tissues more sensitive to insulin action. According to this, studies in animals showed that visceral fat transplanted from GHRKO mice into normal mice enhanced insulin sensitivity and glucose tolerance in the normal mice [24]. In addition, it was demonstrated that the surgical removal of the intra-abdominal fat improves IS in normal animals, but promotes insulin resistance in GHRKO mice, which are characterized by high insulin sensitivity and an extended lifespan [25,26]. These data clearly showed the different physiological roles played by visceral adipose tissue from these animals, indicating that the increase of fat deposit could be associated with beneficial effects on insulin signaling. A possible explanation could be that the cytokines secreted by visceral fat in these animals have an anti-inflammatory rather than pro-inflammatory profile, including an increased expression of adiponectin and a reduced expression of IL-6 and TNFa [27].

In humans, studies were reported in patients from Itabaianinha in Brazil, which have GHD due to a mutation in the GHRH receptor, and in patients from Ecuador and Israel, which have GH resistance, being that all of them were born with impaired GH action. These two first populations have an increase in fatty mass associated with a lower prevalence of type 2 diabetes mellitus (DM 2), reinforcing the hypothesis that the fatty tissue of humans who have a deficiency in GH action since childhood is more sensitive to insulin than that of individuals with normal pituitary function [9,10].

In these three populations, the results of IS are inconsistent and the methods used for IS evaluation could justify these differences. IS was assessed by the glucose tolerance test and HOMA-IR that suffer interference from insulin counter-regulatory hormones. EHC is not affected by the absence of the GH action, an insulin counter-regulatory hormone, since the hyperinsulinemic state during EHC induces the suppression of endogenous glucose production and the pancreatic secretion of insulin, which both may be impaired in patients with GHD [13]. In our study, we showed a higher IS in patients with COGHD using the EHC that faithfully represents IS in these patients.

The results of the Israel cohort are similar to the cohorts of Brazil and Ecuador in relation to the higher PFM. The determination of nutritional intake and measurement of resting energy expenditure by indirect calorimetry in the Israel cohort showed that the increase in the fatty mass, characteristic of these patients, is not due to an increased caloric intake or reduced energy expenditure, thus showing that the fat from the absence of GH action since childhood does not depend on lifestyle. In this cohort, higher adiponectin levels, despite the increase in the PFM, suggest the hypothesis of a particular fatty tissue in this population with COGHD [28].

Few studies have separately evaluated the metabolic parameters of patients with COGHD and AOGHD. The authors of a review on this subject described that adults with untreated COGHD have significantly lower values for body weight, body mass index, and lean body mass than those with AOGHD—results that we also detected in our patients [29].

In humans, a negative correlation between the PFM and IS is well established in the literature [19]. Similarly, we have found an inverse correlation between PFM and IS in CG and in AOGHD. However, patients with COGHD have not shown this correlation, suggesting that the non-exposure to GH in the early stages of life could generate different glucometabolic characteristics to this fatty tissue. These metabolic differences could justify a study that showed no evidence of premature atherosclerosis in Brazilian adult patients with congenital GHD, associated with an increased cardiovascular risk after GH replacement [30,31]. It is worth mentioning that this population without GH replacement has higher FPM [32].

The understanding of the relationship between GH and IS has evolved significantly [33]. Based on our results, we believe that future investigations seeking to better understand glucose metabolism in patients with COGHD may provide benefits for the treatment of diseases related to insulin resistance, such as type 2 diabetes and obesity.

A limitation of this study was the relatively small sample size, due to GHD being a rare disease and also to the strict inclusion criteria to select our patients to perform EHC, a complex method to assess IS. Besides that, the use of electrical bioimpedanciometry as a method for the assessment of body composition can be considered as another limitation of the study. However, this method is well accepted in studies of body composition assessment because it has a good correlation with DEXA (dual-energy X-ray absorptiometry), the gold standard method [34].

It is noteworthy that our two groups of patients with hypopituitarism performed the same replacement regimen and the comparation of IS to the respective CG were different in groups COGHD and AOGHD. In this sense, we consider that the replacement of these hormones did not impact our results, suggesting the presence of another factor influencing the IS than the hormone replacement scheme.

In addition to the gold standard method used to assess IS, comparing the results of two groups of patients according to the stage of disease onset with two control groups matched for age, sex, and BMI makes our data more reliable.

## 5. Conclusions

When compared with their respective control groups, patients with COGHD were more sensitive to the insulin action than paired healthy individuals, while patients with AOGHD showed similar IS. The presence of higher IS, despite higher PFM in patients with COGHD, and the absence of a negative correlation between PFM and IS in these patients suggest that the lack of GH early in life could modify the characteristics of tissues responsible for glucose metabolism. In addition, this change could induce beneficial effects on IS, which remain in the adult life of these patients, showing that the glucometabolic findings of assessments in patients with AOGHD are not applicable for COGHD.

Further studies are needed to confirm our findings and to help the understanding of the mechanisms involved in glucose metabolism in COGHD patients.

## Figures and Tables

**Figure 1 metabolites-12-01251-f001:**
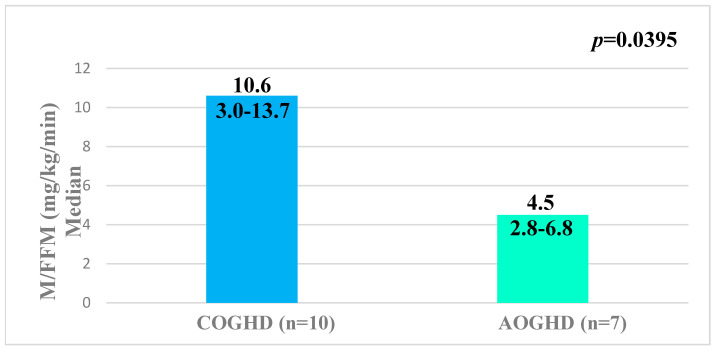
Insulin sensitivity in patients with COGHD vs. AOGHD, COGHD vs. control subjects COGHD and AOGHD vs. control subjects AOGHD. M/FFM was adjusted for age and BMI. Abbreviations: COGHD: patients with childhood-onset GH deficiency; AOGHD: patients with adulthood-onset GH deficiency; M/FFM: M-value normalized per kg free fat mass. The data refer to the median, minimum, and maximum.

**Figure 2 metabolites-12-01251-f002:**
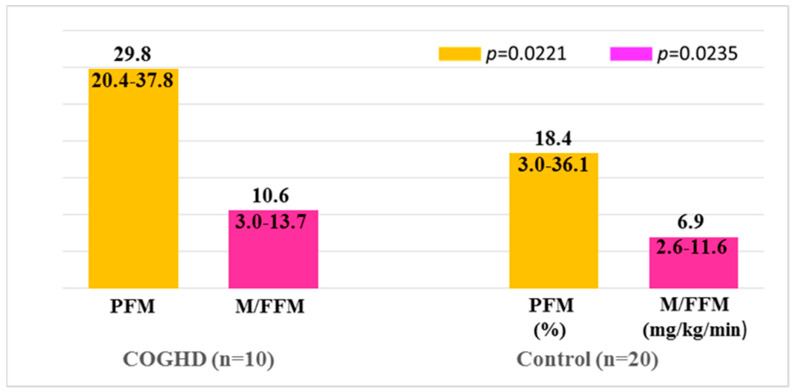
Comparison of PFM and IS (expressed as M/FFM) in patients with COGHD vs. controls. The data refer to the median, minimum, and maximum.

**Figure 3 metabolites-12-01251-f003:**
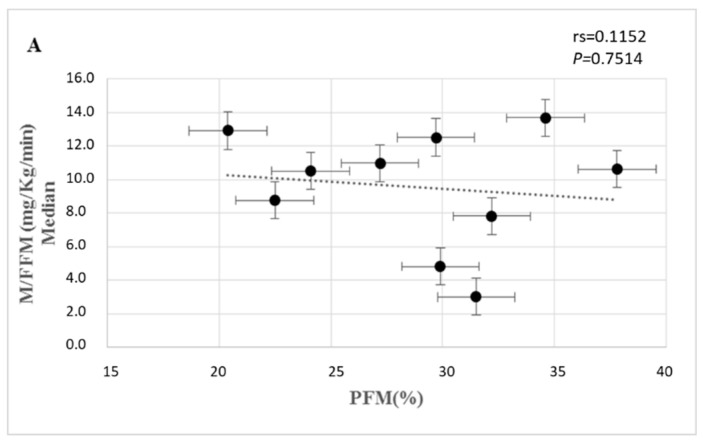
Correlation between insulin sensitivity and percentage of fat mass in patients with COGHD (**A**), patients with AOGHD (**B**), and all control subjects (**C**). Abbreviations: rs: Spearman correlation coefficient.

**Table 1 metabolites-12-01251-t001:** Clinical characteristics, replacement therapy, etiology, laboratory findings, body composition, and insulin sensitivity parameters of each patient with childhood-onset and adult-onset GHD.

Patients (*n* = 17)	Sex	Age (years)	BMI (kg/m^2^)	Treatment	Etiology Hp	IGF-1 (ng/mL)	F-Gly (mg/dL)	PFFM (%)	PFM (%)	M/FFM (mg/kg/min)
Childhood-onset GHD										
Pt-1	M	40	18.80	P-L-T	CPH	34.44	69.7	75.9	24.1	10.5
Pt-2	M	32	24.61	P-L-T-D	CPGP	52.6	77.5	77.5	22.5	8.8
Pt-3	M	23	17.78	P-L-T	CPH	40.1	79.5	62.2	37.8	10.6
Pt-4	M	24	17.70	P-L-T-D	CPGP	18.39	71.9	65.4	34.6	13.7
Pt-5	M	25	20.28	P-L-T	CPH	36.33	77.5	79.6	20.4	12.9
Pt-6	M	22	35.10	P-L-T-D	CPGP	69.65	81.5	68.5	31.5	3.0
Pt-7	F	24	21.52	P-L-Es/Pg	CPH	25.4	75.1	70.3	29.7	12.5
Pt-8	F	22	23.14	P-L-Es/Pg	CPH	36.69	70.7	70.1	29.9	4.8
Pt-9	F	22	19.80	P-L-Es/Pg	CPH	71.68	85.0	67.8	32.2	7.8
Pt-10	M	22	25.30	P-L-T	CPH	70.1	77.4	72.8	27.2	11.0
Adulthood-onset GHD										
Pt-1	F	32	22.06	P-L-Es/Pg	ES	22.67	70.8	65.4	34.6	6.0
Pt-2	M	50	31.92	P-L-T	NFPM	18.8	84.6	65.7	34.3	4.9
Pt-3	F	56	41.21	P-L	SHS	47.64	92.5	56.6	43.4	2.8
Pt-4	M	50	31.50	P-L-T	NFPM	31	71.2	76.2	23.8	6.8
Pt-5	M	41	46.00	P-L-T-D	NFPM	30.11	92.5	63.5	36.5	2.9
Pt-6	F	42	21.90	P-L-Es/Pg	ES	31.5	74.7	66.3	33.7	4.5
Pt-7	F	38	35.2	P-L-Es/Pg	SHS	25.2	78.1	58.1	41.9	2.9

Abbreviations: GHD: growth hormone deficiency; Pt: patient; P: prednisone; L: levothyroxine; Es/Pg: estrogen/progesterone; D: desmopressin; CPH: congenital pituitary hypoplasia; CPGP: craniopharyngioma postsurgical; ES: empty sella; NFPM: non-functioning pituitary macroadenoma.

**Table 2 metabolites-12-01251-t002:** Comparative analysis of clinical, laboratory characteristics and body composition between patients with childhood-onset, adulthood-onset GHD, and their respective control subjects.

Variable	COGHD (*n* = 10)	Control-COGHD (*n* = 20)	AOGHD (*n* = 7)	Control-AOGHD (*n* = 14)
**Sex (female)**	3 (33.3%)	6 (33.3%)	4 (57.1%)	8 (57.1%)
**Age (years)**	28.3 ± 9.6 24.0 (22.0–50.0) ^a^	28.5 ± 8.9 25·5 (20.0–50.0)	40.3 ± 10.9 41·0 (23.0–56.0)	39.0 ± 9.8 39·5 (23.0–54.0)
**BMI (kg/m^2^)**	22.4 ± 5.2 20.9 (17.7–35.1) ^a^	23.4 ± 3.7 23·1 (17.9–34.9)	32.8 ± 9.0 31.9 (21.9–46.0)	33.5 ± 8.6 31.5 (22.6–48.6)
**F-Gly (mg/dL)**	76.6 ± 5.1 77.5 (69.7–85.0) ^b^	86.5 ± 5.9 86.3 (73.3–97.4)	80.6 ± 9.3 78.1 (70.8–92.5)	85.3 ± 7.4 87.9 (73.2–94.6)
**IGF-1 (ng/mL)**	42.8 ± 18.4 36.7 (18.4–71.7) ^b^	195.7 ± 56.6 212.7 (102.4–286.1)	31.1 ± 9.4 30.6 (18.8–47.6) ^c^	133.7 ± 70.7 111.3 (75.4–261.7)
**PFFM (%)**	71.0 ± 5.5 70.2 (62.2–79.6) ^b^	80.8 ± 7.8 81.6 (63.9–97.0)	64.5 ± 6.4 65.4 (56.6–76.2)	65.6 ± 8.6 67.8 (47.2–77.7)
**PFM (%)**	29.0 ± 5.5 29.8 (20.4–37.8) ^b^	19.2 ± 7.8 18.4 (3.0–36.1)	35.5 ± 6.4 34.6 (23.8–43.4)	34.4 ± 8.6 32.3 (22.3–52.8)

Values are shown as mean (standard deviation), median (min-max), or number and frequency (percentage). Abbreviations: GHD: growth hormone deficiency; COGHD: patients with childhood-onset GH deficiency; AOGHD: patients with adulthood-onset GH deficiency; BMI: body mass index; F-Gly: fasting glycemia; PFM: percentage of fat mass; PFFM: percentage of free fat mass. ^a^ *p* < 0.05 (COGHD vs. AOGHD); ^b^ *p* < 0.05 (COGHD vs. control-COGHD), ^c^ *p* < 0.05 (AOGHD vs. control-AOGHD).

## Data Availability

The collected data for this study are in a database, together with the statistical analysis that was carried out by the specialized statistical center of our institution. Patients are identified by number. In addition to this database, the research protocol and the informed consent form are available. Due to ethical and privacy aspects, the data may be requested by email to the corresponding author and a signed data access agreement will be necessary.

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
