# Peer review of "Childhood-Onset GH Deficiency versus Adult-Onset GH Deficiency: Relevant Differences Regarding Insulin Sensitivity"

_metabolites, 2022, doi:10.3390/metabo12121251_

Round 1
Reviewer 1 Report
The paper is devoted to a very interesting topic of insulin sensitivity in the patients with GH deficiency. The study is very interesting and the results are significant from both scientific and clinical point of view. Methodology of the study is appropriate and described clearly. The Discussion is concise but very interesting and impressive; unfortunately, there is only one citation of the paper published in 2020 and none from 2021-2022, so, I would like to encourage the Authors to review the most recent literature once more for the papers that could be included in Discussion. In my opinion the paper is suitable for publication in “Metabolites” after minor but significant revision. I’ve found some inaccuracies that should be corrected or explained, however they do not detract from the substantive value of the presented results. Please see the details in the attached file.
Kind regards, Reviewer

Author Response
Reviewer 1
The paper is devoted to a very interesting topic of insulin sensitivity in the patients with GH deficiency. The study is very interesting and the results are significant from both scientific and clinical point of view. Methodology of the study is appropriate and described clearly. The Discussion is concise but very interesting and impressive; unfortunately, there is only one citation of the paper published in 2020 and none from 2021-2022, so, I would like to encourage the Authors to review the most recent literature once more for the papers that could be included in Discussion. In my opinion the paper is suitable for publication in “Metabolites” after minor but significant revision. I’ve found some inaccuracies that should be corrected or explained, however they do not detract from the substantive value of the presented results. Please see the details in the attached file.
Response: First of all, we would like to thank the reviewer for the excellent evaluation and all the comments referring to our scientific article.
Unfortunately, we did not find in the literature recent original studies evaluating insulin sensitivity through the clamp in adult patients with GHD. Thus, we decided to include a recent study with control group pared by age ,sex and BMI in the discussion:
The results of studies evaluating IS in patients with GHD are controversial. Most studies do not separate patients with COGHD and AOGHD. Although several authors associate GHD in adults with insulin resistance, this association is not well established in the literature. Conversely, a recent study has shown that adult patients with GHD are more sensitive to insulin than the control group matched for age, sex and BMI through HOMA-IR. In our study, using the HEC, we found no differences in SI between patients with AOGHD and the matched control group.
The following questions should be corrected or clarified before publication:
Please check the abbreviations used in the text and in the Tables – is “FFMP” the same as “PPFM”? It’s difficult to understand the results when the abbreviations are different and this should definitely be unified before publication.
Response: The reviewer is correct, we unified the abbreviations
In my opinion, the word „contradictory” in lines 10 and 34 is used inappropriately, as in the cited paper, it relates to the discrepant results of different studies of the pattern of IS not to this patternper se; similarly the problem of contradictory results of different studies is described in nextsentences by the Authors (in line 41 “contradictory” is used in a proper context).
Response: We agree with the reviewer and changed the text
In line 88, a Reference (or References) concerning the guideline standards applied should be provided.
Response: We provided two references.
The way of writing numbers in text and Figures should be unified, using dots, not commas.
Response: We agree with the reviewer and changed the text
Please, describe the meaning of whiskers in Figure 1 A-C; whiskers on the left are hardly visible – is it possible to prepare color figures? The same relates to visibility of the text on darker box in Figure 2 (p=0.0235???).
Response: We agree with the reviewer and we changed the figures.
The description of Fig. 2 should not be a conclusion but an explanation of the picture, e.g.“Comparison of FMP and IS (expressed as M/FFM) in patients with COGHD vs Controls”, while the statement placed as description of this Figure by the Authors could be placed directly in the main text.
Response: We agree with the reviewer and changed description of Fig. 2
Lines 204-205 – in fact, the Authors have not compared the patients with COGHD and AOGHD with similar body composition but have adjusted the results for BMI – this sentence should be re-written, accordingly (see lines 210-211 and not duplicate the same information).
Response: We agree with the reviewer and re-wrote this sentence:
It is noteworthy that, initially, our patients were selected for evaluation of IS regardless of the stage of disease onset for a previously published study [13] . Subsequently, it was decided to divide them into two groups: COGHD and AOGHD, groups not matched for BMI and age. The direct comparison between these two different groups is not ideal, but the influence of the lower BMI and age found in patients with COGHD on IS results was statistically compensated with adjustments for age and BMI.
Line 213 – what is the meaning of References [17,18] in the context of presenting own results only? These References should be explained or removed.
Response: The reviewer is correct. We changed the place of these references
Some minor editorial errors should be corrected:
Please fill in missing spaces, especially between abbreviations and brackets, e.g.
Lines 16, 17, 19 – between abbreviations and brackets
Line 19 – would be easier to read: “FMP (rs: -0.8214, p=0.0234 and rs: -0.3639, p<0.0344)”
Line 20 – missing spaces similarly as in line 19
Line 86 – missing spaces between “1” and “mg”, as well as between “2” and “mg”
Line 155 – missing space between “IGF-1 and”
Line 162 – missing space between “BMI:” and “Body”
Line 226 – missing space between “[20].” and “In”
Please remove also redundant (double) spaces in the text, e.g in line 34, line 143, line 162, line 175, line 219
Line 66 – abbreviation “FMP” should be introduced (not only FFMP)
Line 156 – bracket opened in the end of line, please move to line 157
Line 157 – should be “(Table 2 and Fig. 1A)”
Line 70 – should be “healthy individuals”
Line 80 – it would be better to write “free thyroxine (FT4)”
Lines 81-82 – the abbreviation “IGF-1” should transferred to line 82
Line 97 – should be “healthy individuals”
Line 107 – the abbreviation “BC” should be introduced when used for the first time
Line 110 – should be “The subjects”
Lines 141-142 – the abbreviation “IGF-1” should transferred to line 82
First line in Table 1 – shouldn’t be “Patients (n=10)”, please explain if “9” is appropriate
Line 176 – should be “in patients with COGHD than in controls”
Line 216 – IS may be “higher than” or “similar to” – please, re-write the sentence
Line 224 – it should be better “studies in animals” (I suppose it was not a single animal studied)
Line 242 – please introduce the abbreviation “GTT” or write only the full name of the test, if the
abbreviation is not used anywhere else
Line 255 – shouldn’t be “reinforce” if this refers to “levels”
Line 263 – should better be “cardio-vascular” (or “cardiovascular”)
Response: We agree with the reviewer and have done all requested correction.
Please, check if Reference list is prepared according to the Template, it’s done differently than in the papers published recently in “Metabolites”; doi numbers should be provided where possible
Response: We agree with the reviewer and have made the requested correction in the Reference.
Line 365 – should all the title of the paper be written in capital letters?
Response: We agree with the reviewer and have made the requested correction.
Please see the attachment with the complete manuscript
Reviewer 2 Report
Manuscript ID: metabolites-2032864
Title: Childhood-onset GH deficiency versus Adult-onset GH deficiency: relevant differences regarding insulin sensitivity.
In this interesting study the authors assess insulin sensitivity in GH deficient adults.
The two GHD groups (COGHD and AOGHD) are very different (age, BMI, fat mass as some examples) so the comparison between these two groups is not ideal, even when adjusted for these parameters.
Since GH had direct metabolic effects, the fact that both GHD groups were not receiving GH should be discussed when they are compared to the controls.
Figure 1 a, b and c could be combined as one. No need to present as 3.
Page 7 line 185: the authors state: “Only levels of IGF-1 were significantly lower in patients with AOGHD (Table 2).” Table 2 does not have IGF-I data.
In the discussion the authors need to be careful with the comparisons. The population from Itabaianinha has GHD due to mutation in the GHRH-R. The populations from Ecuador and from Israel have GH resistance and not GHD.
Author Response
Reviewer 2
The authors compared insulin sensitivity in childhood-onset GH deficiency VS adult-onset GH deficiency using euglycemic hyperinsulinemic clamp. I have the following suggestion/comments:
Response: First, we thank the reviewer for evaluating our study.
Line 75: in patient section: please provide more detailed information about patients with childhood-onset GH deficiency including age onset, if patients were treated with GH during childhood, duration of treatment with GH, and duration for the time of stopping GH treatment to the time of study.
Response: We collect this necessary data and include this data in the text:
These patients were not under GH replacement, because this replacement in adult patients with GHD was not routinely performed in our outpatient clinic. Patients with COGHD were aged 6 (±1.16) years at diagnosis. All these patients received GH replacement, the mean treatment time was 10 (±1.11) years and the treatment interruption time before the start of the study was 5 (±1.35) years.
Line 98: why control group in childhood-onset GH deficiency was not matched for FFM index or PFFM instead of BMI?
Response: Thanks for the interesting question. We performed the body composition assessment only after selecting patients and controls based on clinical data, so the control group was matched by BMI, age and sex. In our protocol, body composition was established as an outcome of our study
Figure 1: Please indicate the cut off M-value for insulin resistance. Need to determine the proportion of patients with insulin resistance and compare among the difference groups.
Response: The main objective of our study was to assess insulin sensitivity in the context of GH deficiency and to analyze the possible influence of the stage of disease onset on this metabolic state. In this sense, categorizing patients according to the presence of insulin resistance could affect our analysis due to the small size of the population studied and would not allow us to fulfill our main objective. Another important aspect for not using this division is that there is no consensus in the literature on the cutoff values of the M-value that define insulin resistance. In these studies, it is considered an arbitrary division due to the influence of many variables. For these reasons, we decided not to divide the patients according to whether or not they had insulin resistance.
- Tam CS, Xie W, Johnson WD, Cefalu WT, Redman LM, Ravussin E. Defining insulin resistance from hyperinsulinemic-euglycemic clamps.Diabetes Care. 2012; 35 (7): 1605–10.
- Bergman RN, Finegood DT, Ader M. Assessment of Insulin Sensitivity in Vivo. Endocr Reviews. 1985; 6 (1): 45–86.
- Tuzcu A, Yalaki S, Arikan S, Gokalp D, Bahcec M, Tuzcu S. Evaluation of insulin sensitivity in hyperprolactinemic subjects by euglycemic hyperinsulinemic clamp technique. Pituitary;12 (4): 330–4.
- Stern SE, Williams K, Ferrannini E, DeFronzo RA, Bogardus C, Stern MP. Identification of Individuals With Insulin Resistance Using Routine Clinical Measurements. Diabetes. 2005; 54 (2): 333–9.
Line 200: Please explain why patients with childhood-onset GH deficiency had much lower BMI than those with adult-onset.
Response: Your questioning about differences between groups of patients alerted us to the need to explain how our work was carried out, in stages.Our first work was to evaluate insulin sensitivity in a group of patients with GHD using the EHC method. In carrying out this evaluation, we found that COGHD patients were much more insulin sensitive than AOGHD patients. Based on this finding, we decided to divide our patients with GHD into two groups that, therefore, were not matched. To minimize these differences we included two groups matched by age, sex and BMI to compare with each group of patients.Due to the questioning of the reviewer, we decided to include two new phrases in the discussion of our study:
It is noteworthy that, initially, our patients were selected for evaluation of IS regardless of the stage of disease onset for a previously published study [13] . Subsequently, it was decided to divide them into two groups: COGHD and AOGHD, groups not matched for BMI and age.
Few studies have separately evaluated the metabolic parameters of patients with COGHD and AOGHD. The authors of a review on this subject described that adults with untreated COGHD have significantly lower values for body weight, body mass index and lean body mass than those with AOGHD, results that we also detected in our patients [29].
Reviewer 3 Report
The authors compared insulin sensitivity in childhood-onset GH deficiency VS adult-onset GH deficiency using euglycemic hyperinsulinemic clamp. I have the following suggestion/comments:
Line 75: in patient section: please provide more detailed information about patients with childhood-onset GH deficiency including age onset, if patients were treated with GH during childhood, duration of treatment with GH, and duration for the time of stopping GH treatment to the time of study.
Line 98: why control group in childhood-onset GH deficiency was not matched for FFM index or PFFM instead of BMI?
Figure 1: Please indicate the cut off M-value for insulin resistance. Need to determine the proportion of patients with insulin resistance and compare among the difference groups.
Line 200: Please explain why patients with childhood-onset GH deficiency had much lower BMI than those with adult-onset.
Author Response
Reviewer 3
Thank you for the opportunity to review the manuscript “Childhood-Onset GH Deficiency versus Adult-Onset GH Deficiency: Relevant Differences Regarding Insulin Sensitivity” by Heraldo Mendes Garmes et al. It refers to an important issue of IS in hypopituitarism, especially in patients with GHD. I find the paper interesting and valuable, especially for clinicians.
First, we would like to thank the reviewer for evaluating our study.
However, there are some issues that need to be improved.
- Lines 10-11: “Introduction: The pattern of insulin sensitivity (IS) is contradictory (please change style) in patients with GH deficiency (GHD), however, the interference of the GHDonset stage, childhood or adulthood in the IS has not been assessed (please write it in less definitive way) by euglycemic hyperinsulinemic clamp (EHC), a gold-standard method (for?).” – please modify the Introduction.
Response: We agree with the reviewer and changed the text.
- Lines 20-21: “Discussion: (Conclusions) Despite the higher FMP, patients with COGHD were more sensitive to the insulin than paired healthy individuals, while patients with AOGHD showed similar IS. (comparing to controls)”
Response: We agree with the reviewer and changed the text.
- Lines 23-21: Thus, the glycometabolic findings in patients with COGHD are not applicable (seems to be not applicable) to AOGHD.
Response: We agree with the reviewer and changed the text.
- Lines 46-50: “IS HOMA-IR (please add explanation for an abbreviation) and Euglycemic Hyperinsulinemic Clamp (EHC) was poor in patients with hypopituitarism [13]. In addition, it has been reported that standard minimum homeostatic models for assessing IS could not adequately identify IS in patients with COGHD [14]. These findings highlighted the relevance of EHC in the investigation of IS, since the EHC does not suffer interference from the insulin counter-regulatory hormones, which are altered in patients with hypopituitarism”- I am not sure if I understand the paragraph correctly, at first the Authors write that EHC is poor, then that EHC is a relevant investigation of IS.
Response: We agree with the reviewer and added explanation for an abbreviation.
Standard minimum homeostatic models for assessing IS is poor, but EHC is better.
The question was explained in the changed text “Many factors have been reported to explain these contradictory aspects about IS in GHD; such as the etiology of GHD and the influence of other hormones replacement therapy in patients with hypopituitarism [11,12]. Another important element is the method used to assess IS. In this sense, in a previous study, our group showed that, unlike what was observed in individuals with normal pituitary function, the agreement between the assessment of IS by the Homeostatic Model Assessment for Insulin Resistance (HOMA-IR) and the Euglycemic Hyperinsulinemic Clamp (EHC) was poor in patients with hypopituitarism [13]. This was also reported in another study where it is suggested that standard minimal homeostatic models to assess IS such as HOMA-IR could not adequately identify IS in syndromes of severe short stature [14].”
- I would recommend to present clearly the main aims of the study at the end of the Introducion.
Response: We agree with the reviewer and added main aims of the study at the end of the Introducion: For this reason we decided to carry out a study to evaluate insulin sensitivity and body composition in adult patients with GH deficiency according to the stage of disease onset and compare it with a control group paired for age, sex and BMI.
- Lines 63, 84: “All (there is an exception- pt 3 AOGHD) patients were on levothyroxine, prednisone, estrogen and progesterone or testosterone replacement and none on GH therapy. (there is a necessity to explain why they were not on rhGH therapy)”
Response: The reviewer is correct, we wrote about the exception and we explained why they are not on rhGH therapy: These patients were not under GH replacement, because this replacement in adult patients with GHD was not routinely performed in our outpatient clinic.
- Line 81: rather stimulated than “basal cortisol, GH”.
Response: We agree with the reviewer and changed the text
- Line 87: “according to current guideline standards.” – please add reference
Response: Thank you, We put the references,
- Line 93: IRN- please add explanation for an abbreviation.
Response: We agree with the reviewer and added explanation for an abbreviation
- Line 99: “The individuals were recruited among the relatives of patients in the outpatient clinic of our hospital.” – where they relatives of the study pts?
Response: The reviewer is correct, We changed the text: The healthy individuals were recruited among the accompanist of patients in the outpatient clinic of our hospital.
- I would suggest widening the discussion regarding unexpected findings that patients with AOGHD showed similar IS to controls.
Response: We agree with the reviewer and added in the discussion:
The results of studies evaluating IS in patients with GHD are controversial [4]. Most studies do not separate patients with COGHD and AOGHD. Although several authors associate GHD in adults with insulin resistance, this association is not well established in the literature. Conversely, a recent study has shown that adult patients with GHD are more sensitive to insulin than the control group matched for age, sex and BMI through HOMA-IR [22,23]. In our study, using the HEC, we found no differences in SI between patients with AOGHD and the matched control group.
- Lines 286-287: “When compared with their respective control groups, patients with COGHD were more sensitive to the insulin action than paired healthy individuals, while patients with AOGHD showed similar IS.”
Response: We agree with the reviewer and changed the text
Reviewer 4 Report
Please see the attached review.

Author Response
Reviewer 4
Title: Childhood-onset GH deficiency versus Adult-onset GH deficiency:
relevant differences regarding insulin sensitivity.
In this interesting study the authors assess insulin sensitivity in GH
deficient adults.
First, we thank the reviewer for evaluating our study.
The two GHD groups (COGHD and AOGHD) are very different (age, BMI, fat
mass as some examples) so the comparison between these two groups is not
ideal, even when adjusted for these parameters.
Response: We agree with the reviewer, for this reason, we made two matched control groups and the matched results were different for COGHD patients and similar for AOGHD patients when compared to a respective control groups. We changed our discussion and whore a new frase: It is noteworthy that, initially, our patients were selected for evaluation of IS regardless of the stage of disease onset for a previously published study. Subsequently, it was decided to divide them into two groups: COGHD and AOGHD, groups not matched for BMI and age. The direct comparison between these two different groups is not ideal, but the influence of the lower BMI and age found in patients with COGHD on IS results was statistically compensated with adjustments for age and BMI.
Since GH had direct metabolic effects, the fact that both GHD groups
were not receiving GH should be discussed when they are compared to the
controls.
Response: Regarding the direct effects of GH on metabolism, we did not find studies in the literature comparing patients with COGHD and AOGHD with control groups matched by age, gender and BMI. However, we decided to include in the discussion: Few studies have separately evaluated the metabolic parameters of patients with COGHD and AOGHD. The authors of a review on this subject described that adults with untreated COGHD have significantly lower values for body weight, body mass index and lean body mass than those with AOGHD, results that we also detected in our patients
Figure 1 a, b and c could be combined as one. No need to present as 3.
Response: We agree with the reviewer and we changed the text to just figure 1.
Page 7 line 185: the authors state: “Only levels of IGF-1 were
significantly lower in patients with AOGHD (Table 2).” Table 2 does not
have IGF-I data.
Response: The reviewer is correct. We put the IGF-1 in the table 2
In the discussion the authors need to be careful with the comparisons.
The population from Itabaianinha has GHD due to mutation in the GHRH-R.
The populations from Ecuador and from Israel have GH resistance and not GHD.
Response: We agree totally with the reviewer. Although none of them have GH action, one due to lack of GH and the other due to lack of GH receptor, this difference was not clear in the text. Because of this, we put in the discussion:
=population from Itabaianinha, which has GHD due to mutation in the GHRH-R.
=populations from Ecuador and from Israel, which have GH resistance.
Round 2
Reviewer 1 Report
Thank you for for referring to my comments and re-submitting the corrected version of the paper. I've now only several minor corrections that should be regarded before publication (see the attachement).
I hope the paper would be successfully published.
Kind regards, Reviewer

Author Response
Please find below my comments to the current version of the manuscript:
Line 7 and 8-9 – why the same University has different affiliations (exactly: what does “and” before “Brazil” mean?)
Response: We thank the suggestion. We withdrawal the word “and” and included the right affiliation.
Line 10 and line 34 – in my opinion, it should be better re-written once more – the pattern of insulin sensitivity is neither contradictory nor discrepant but the results of the studies on the pattern of insulin sensitivity lead to discrepant results, i.e. the results of the studies on the pattern of insulin sensitivity are contradictory.
Response: We thank the suggestion. We agree and rewrite the sentences.
I do not understand the term “stage of the disease onset” – wouldn’t be better simply “patient’s age at the disease onset” (line 61 and line 218).
Response: We thank the suggestion. We agree and rewrite the sentences.
Line 109 – the word “relatives” has been change to “accompanist” – have you meant “the persons accompanying the patients” – please clarify and use the plural not singular?
Response: We thank the suggestion. We agree and rewrote the sentences.
Unfortunately, in first raw of Table 1 are the abbreviations “PFFM” and PFM, while in lines 70-71 are introduces the abbreviations “FFMP” and “FMP”. The authors have declared this problem as resolved and in Table 2 it’s really OK.
Response: We agree and rewrote to PFFM” and PFM”.
In Table 2 superscripts concerning significant differences should be placed in alphabetical order, while now “c” is before “a” and “b” – should be corrected.
Response: We agree and rewrote in the alphabetical order.
Figures are much better now, however the notation of numbers in the text and in the Figures is still different – please unify.
Response: The reviewer is correct. We unified the notation of numbers.
Figure 1 should be slightly reduced in size so that the entire description under the Figure fits on the same page. I understand that bars in this Figure represent Medians, unfortunately, the meaning of whiskers is still not explained.
Response: We reduced the size of the figures and fitted on the same page and we state that the data in figure 1 refer to the median, minimum and maximum value.
Figure 2 repeats partially one graph in Figure 1 (the same M/FFM are presented). I suppose that these are Medians as in Figure 1 but the description is missing and should be placed. Moreover only this data is commented in text, while there is no direct reference to Figure 2 in comments concerning
Response: We agree with the reviewer and we state that the data in figure 2 refer to the median, minimum and maximum. The figure 2 ir referred in the manuscript lines 193 and 194.
FMP (%). Information concerning statistical significance of differences illustrated in Fig. 2 should be added (as it’s done in Fig. 1). Why the authors have not shown similar Figure for AOHD?
Response: We agree with the reviewer and we complete figure 2 with the value of statistical significance.
We did not include data from patients with AOGHD in Figure 2, as there was no statistical difference (it is shown in Table 2). In Figure 2, we want to show data with statistical difference and emphasize that patients with COGHD, despite having a higher percentage of fat mass, have greater insulin sensitivity, a relationship that is not seen in the general population.
Line 207 – one of the two dots et the end of the sentence should be removed.
Response: We removed one dot. Thank you for the observation.
Line 208 – please start the section Discussion from the top of next page.
Response: We did it. Thank you for the observation.
Reviewer 3 Report
I accept the revised version in present form.
Author Response
We thank reviewer for the very good review.